

# SigmoID: a user-friendly tool for improving bacterial genome annotation through analysis of transcription control signals

Yevgeny Nikolaichik and Aliaksandr U. Damienikan

Department of Molecular Biology, Belarusian State University, Minsk, Belarus

## ABSTRACT

The majority of bacterial genome annotations are currently automated and based on a 'gene by gene' approach. Regulatory signals and operon structures are rarely taken into account which often results in incomplete and even incorrect gene function assignments. Here we present SigmoID, a cross-platform (OS X, Linux and Windows) open-source application aiming at simplifying the identification of transcription regulatory sites (promoters, transcription factor binding sites and terminators) in bacterial genomes and providing assistance in correcting annotations in accordance with regulatory information. SigmoID combines a user-friendly graphical interface to well known command line tools with a genome browser for visualising regulatory elements in genomic context. Integrated access to online databases with regulatory information (RegPrecise and RegulonDB) and web-based search engines speeds up genome analysis and simplifies correction of genome annotation. We demonstrate some features of SigmoID by constructing a series of regulatory protein binding site profiles for two groups of bacteria: Soft Rot *Enterobacteriaceae* (*Pectobacterium* and *Dickeya* spp.) and *Pseudomonas* spp. Furthermore, we inferred over 900 transcription factor binding sites and alternative sigma factor promoters in the annotated genome of *Pectobacterium atrosepticum*. These regulatory signals control putative transcription units covering about 40% of the *P. atrosepticum* chromosome. Reviewing the annotation in cases where it didn't fit with regulatory information allowed us to correct product and gene names for over 300 loci.

# INTRODUCTION

With bacterial genome sequencing becoming easily accessible, the demand for quality genome annotation has increased drastically in recent years. Manual genome annotation requires a lot of effort and is time-consuming. Therefore, it is usually beyond the scope of most sequencing projects. As a result, automated approaches are commonly used by researchers while submitting genome sequences to GenBank. The genome annotation pipelines such as Prokka, BASYS, RAST and NCBI Prokaryotic Genome Annotation Pipeline (*Van Domselaar et al., 2005*; *Aziz et al., 2008*; *Tatusova et al., 2013*;

Corresponding author
Yevgeny Nikolaichik,
nikolaichik@bio.bsu.by

*Seemann, 2014*) find positions and infer possible functions for most protein and rRNA/tRNA coding sequences, but they are less efficient at finding other structural elements of genome. In particular, none of the popular annotation pipelines cited above identifies regulatory elements such as promoters, transcriptional terminators and transcription factor binding sites (TFBS). However, regulatory elements provide important information about physiological conditions that influence gene activity. Therefore, information about the non-coding sequences of a gene might improve assignment of the gene's function.

Promoters/TFBS and terminators also mark operon borders. Genes within an operon are usually functionally connected and operon structures are normally conserved in related species. Hence, prediction of transcription termination sites, combined with prediction of operon beginnings (marked by promoters and/or TFBSs) gives a good idea about possible operon structure and can help to further improve the quality of functional annotation.

A lot of experimental data on gene regulation by various transcription factors and RNA polymerase sigma factors has been accumulated in recent years, especially on model organisms. A large portion of regulatory information is conveniently accessible from several databases like RegPrecise, RegulonDB and RegTransBase (*Novichkov et al., 2013*; *Gama-Castro et al., 2010*; *Cipriano et al., 2013*; *Kılıç et al., 2013*).

Many existing software packages handle the task of identifying regulatory protein binding sites in DNA sequences. These applications usually rely on binding site representation in the form of positional weight matrix and operate in pair with a tool generating these matrices in the required format. Since their first appearance decades ago, PatSer/Consensus (*Stormo et al., 1982*; *Hertz, Hartzell & Stormo, 1990*), ScanAce/AlignAce (*Hughes et al., 2000*) and MAST/MEME (*Bailey & Elkan, 1994*; *Bailey & Gribskov, 1998*) have been actively used for this purpose.

The vast majority of "traditional" TFBS identification packages do not take into account genome annotation when performing the searches and produce lists of the resulting hit coordinates in the plain text format. The end user then has to manually relate these hit coordinates to the gene/CDS positions to find the corresponding genes. This process becomes inconvenient, cumbersome and slow when many regulators have to be analysed on a whole genome scale. Some GUI tools, including GenomeExplorer (*Mironov, Vinokurova & Gel'fand, 2000*), RegPredict (*Novichkov et al., 2010b*) and Virtual Footprint (*Grote et al., 2009*) simplify this task by displaying the identified TFBSs in genomic context. However, to our knowledge neither of the available TFBS/promoter identification software tools can be used for editing genome annotation. On the other hand, genome browsers like Artemis (*Carver et al., 2012*) are specially designed for editing genome annotation, but none of these browsers can identify regulatory sequences. At the moment, improving genome annotation in accordance with regulatory information requires the use of separate tools for regulatory sequence identification and editing genome annotation, which makes the process laborious and slow. Hence, a tool automating this process would speed it up and improve its consistency and reproducibility.

We developed an application that combines a user-friendly graphical interface to command line tools for regulatory elements identification with a genome browser that has database search and genome editing abilities. Such a combination of features allows

to identify annotation inconsistencies and provides an opportunity to correct errors more easily. Another purpose of developing SigmoID was to create a collection of TFBS/promoter search profiles for plant pathogens from the genus *Pectobacteria* and its close relatives from the genus *Dickeya* to facilitate the studies on plant-pathogen interactions involving this important group of pathogenic bacteria. As a step in this direction and to illustrate the ways SigmoID could be utilised, we have updated the annotation of the recently sequenced *Pectobacterium atrosepticum* genome (*Nikolaichik et al., 2014*) by including the information on regulatory sequences while correcting many annotation errors along the way.

## METHODS AND DATA

### Implementation

SigmoID front end is a GUI application written in Xojo (http://xojo.com) which gives it the expected look and usability on all three supported platforms (OS X, Linux and Windows). The GUI displays TFBS in the form of sequence logos (*Schneider & Stephens, 1990*) and provides an interface for three regulatory sequence search engines: nhmmer (*Wheeler & Eddy, 2013*) from the HMMER package (*Eddy, 2011*), MAST (*Bailey & Gribskov, 1998*) from the MEME Suite (*Bailey et al., 2015*) and TransTerm HP (*Kingsford, Ayanbule & Salzberg, 2007*). SigmoID has configuration windows (Fig. S1) for each of the included non-GUI tools. These windows allow to set search parameters and provide default values where possible. Processing of nhmmer, MAST and TransTerm HP outputs, sequence format conversions and addition of regulatory sites to genome annotations are implemented via Python scripts. The scripts are using Biopython modules (*Cock et al., 2009*), hence Python (2.7.x) and BioPython (version 1.64 and above) are required. The scripts are called from the GUI, but could be used separately and integrated into an annotation pipeline if desired. Detailed installation instructions are provided within distributions for each platform. The source code for the whole SigmoID application is available with GPL 3.0 licence (https://github.com/nikolaichik/SigmoID) and is also archived at Zenodo (http://dx.doi.org/10.5281/zenodo.48831). The compiled applications for all three platforms are also available from the same GitHub repository.

### Basic usage

Several scenarios of SigmoID usage are briefly described below while additional documentation is provided with the distribution.

#### Genome search for TFBS/promoters

Genome search in SigmoID requires only a text file in the FASTA format with aligned sequences for a transcription regulator. The required data can be extracted from the literature or one of the several databases. SigmoID provides an integrated online access to the RegPrecise database (*Novichkov et al., 2013*) which has a large collection of data on transcription factors from a growing number of organisms. Online access is also provided to RegulonDB, the database containing information on most regulators characterised for *Escherichia coli* (*Salgado et al., 2013*). Genome search can also be performed with pre-calibrated TFBS/promoter profiles specific for a certain group of bacteria.

Pre-calibrated profiles can significantly speed up analysis of newly sequenced genomes of bacteria from the same or closely related taxa. The current version of SigmoID includes two sets of such TFBS/promoter profiles optimised for (i) Soft Rot *Enterobacteriaceae* (SRE) including *Pectobacterium* and *Dickeya* genera and (ii) *Pseudomonas* spp. Currently active profile set can be changed in SigmoID preferences by switching to another profile folder distributed with SigmoID or provided by user.

Once the TFBS/promoter data are loaded and displayed in the main window, one of the two algorithms (nhmmer or MAST) can be used for searching genome sequences in GenBank format. The search results are displayed in the main window in the plain text format. SigmoID can add newly discovered regulatory elements to the annotation. At this stage the search results are filtered to remove sites with the wrong orientation relatively to the closest ORFs, sites located downstream of converging ORFs, and (as an option) sites within ORFs. In addition, because nhmmer reports hits on both strands, one duplicate for the palindromic TFBSs is removed. This processing is done by python scripts (separate for each search algorithm) that can be called via the "Annotate Current Sites…" command. The output is saved in a separate GenBank file which is opened in the integrated genome browser to present filtered results visually in the genomic context. Search thresholds and filtering options can be set by the user via the configuration windows (Figs. S1A and S1B); calibrated profiles have all required settings pre-configured.

### Construction of optimised TFBS/promoter profiles

Utility functions within SigmoID are provided for extension/shortening/partial masking of an existing alignment of TFBS/promoter sequences. Displaying an alignment sequence logo helps to choose correct boundaries of the site (Fig. 1). Genome browser provides an opportunity to display nhmmer/MAST search hits in the genomic context; the newly discovered regulatory sites could then be copied or saved in a new file. Finally, a calibrated profile could be saved with the help of the Profile Wizard window (Fig. S1C). Calibration thresholds have to be set by the user, but choosing the right calibration values is simplified with a function that identifies the minimal score for individual sequences in the training set. The cut-off values are stored in the final calibrated hidden Markov model (HMM) profile produced by hmmbuild and are applied by default for further searches with the stored profile. For MAST searches, the profiles are converted into position specific scoring matrices and stored in the MEME format. For these matrices, a single *p*-value threshold should be set by the user. Alongside the calibrated profiles, SigmoID also stores the settings for running nhmmer/MAST and the post-processing scripts as well as the description of the profile.

### Transcription terminator search

This function provides a simple GUI for configuring TransTerm HP, performs necessary sequence format conversions and adds transcriptional terminator annotations to genome files in the GenBank format. The configuration window (Fig. S1D) allows setting TransTerm confidence score and terminator stem/loop parameters. The default values provided should be appropriate to use in most cases.

This function can also be called during the genome scan.

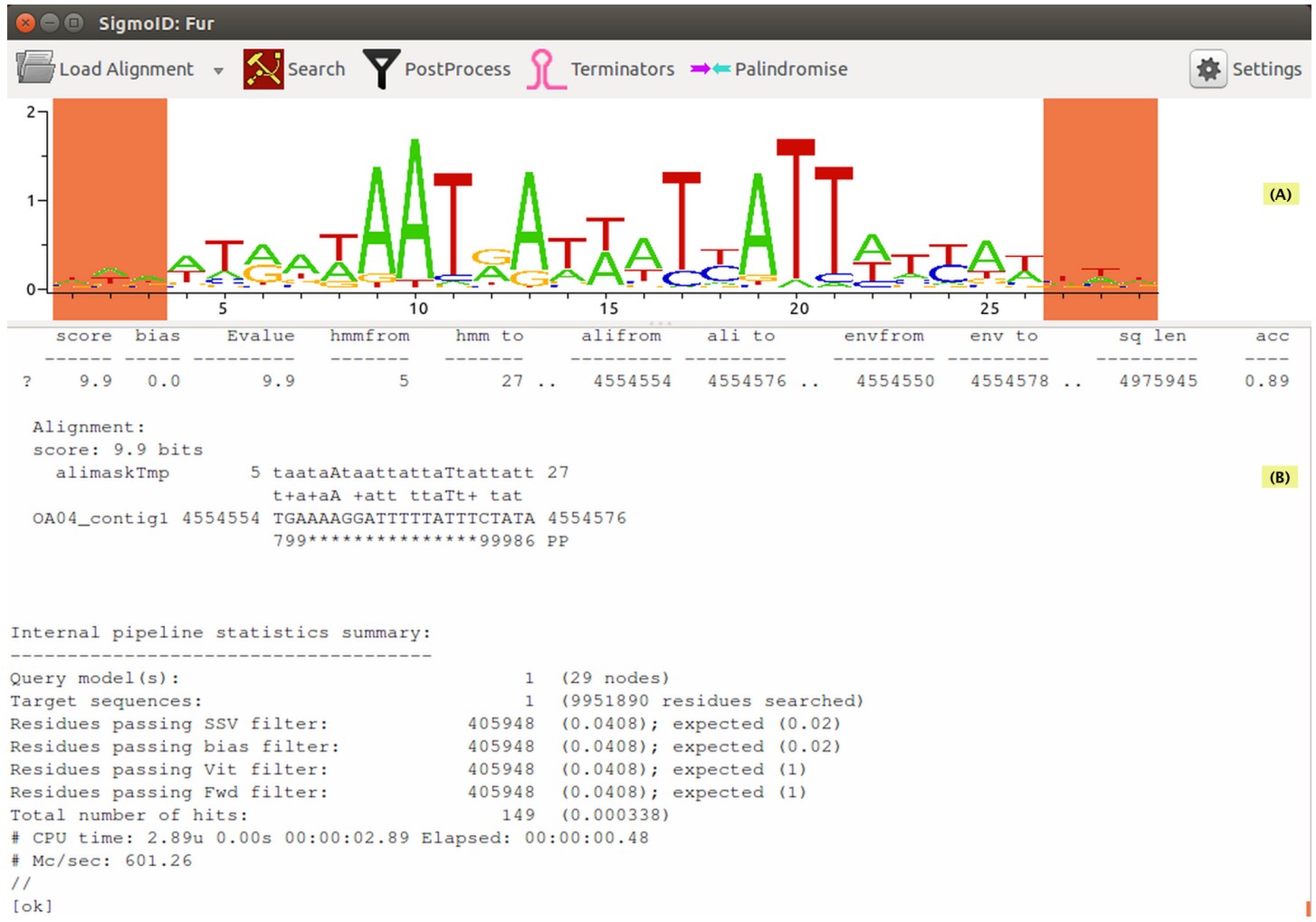

**Figure 1  The main SigmoID window.** The window is divided into two panes: (A) and (B). (A) shows the various pieces of information about the currently opened alignment of TFBS sequences: the sequence logo (on this figure), raw sequences, alignment description, profile data and search settings. Dragging across the logo selects part of the alignment which can be exported in FASTA format; shift-dragging can select several parts of the alignment. Starting a search with part(s) of the logo highlighted (as shown here) masks the highlighted positions with alimask and then performs a nhmmer search with the masked profile. (B) displays standard output from the search programmes and various information messages. The very end of nhmmer output is shown.

### Genome scan

The Genome Scan function searches a chosen genome TFBS/promoters with a number (or all) of the provided profiles (or calibrated profiles created by the user) and transcriptional terminators and adds corresponding annotations to the GenBank file. This function can only be used with the pre-calibrated HMM profiles. nhmmer is used as the search engine with this function and the corresponding post-processing script is run with the settings stored in the calibrated profiles.

### Editing genome annotation

Genome browser (Fig. 2) could be opened after a search to skim quickly through the newly found sites and inspect their genomic context. Running a terminator search beforehand

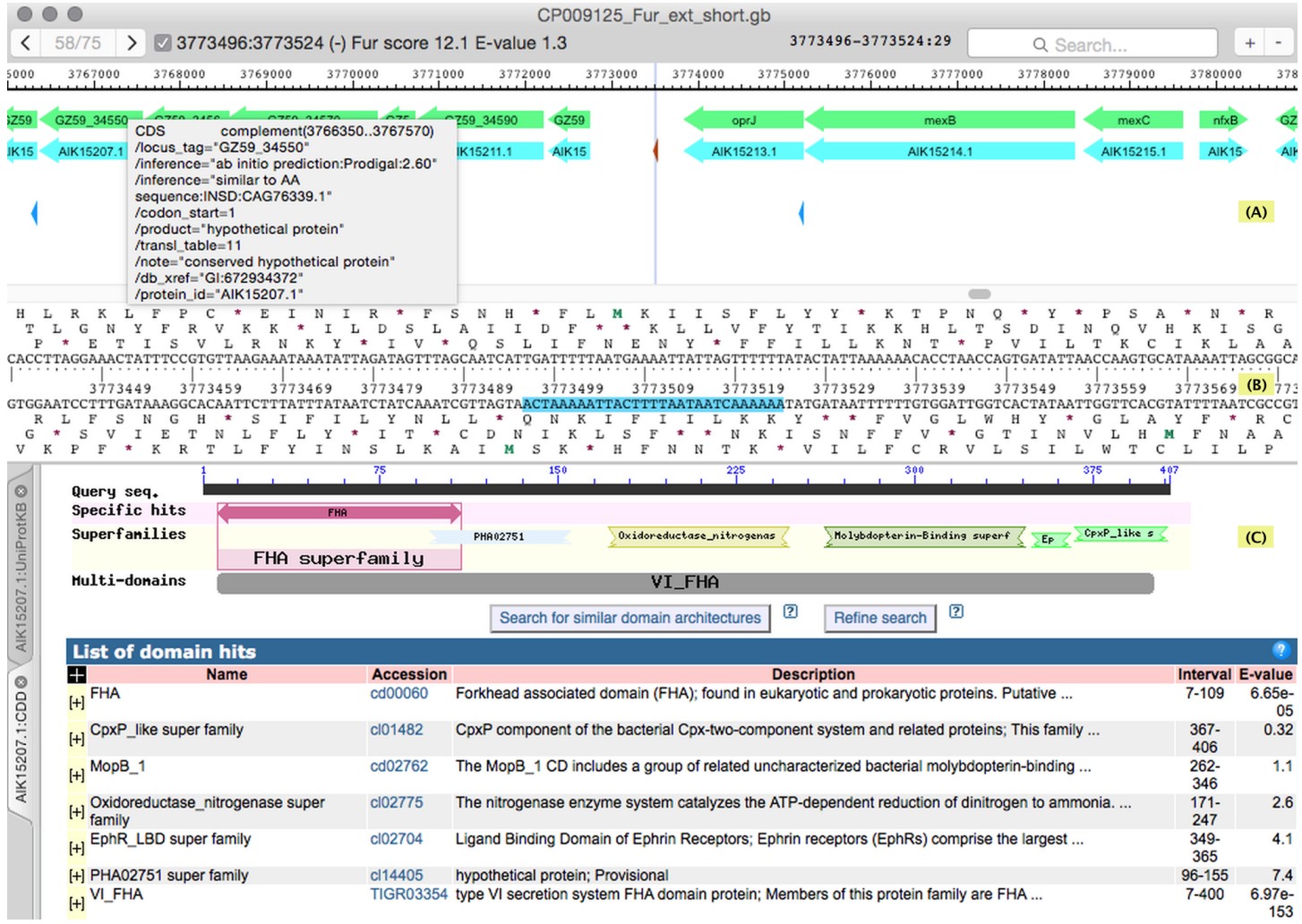

**Figure 2   The genome browser window.** The window is divided into three panes: (A), (B), (C). (A) shows the map of the annotated features, (B)—nucleotide/amino acid sequence of the currently highlighted part of the feature map, and the bottom one—the results of database searches. (C) is a tabbed web browser with navigation commands available via contextual menus. This allows to investigate the details of a particular search hit quickly. Hovering the mouse pointer over a feature in the top pane pops up a small window displaying its current annotation (shown). Clicking on a feature selects it, right-clicking displays a contextual menu allowing to launch a number of web-based searches with a selected feature as a query; the menu also has commands for feature copying and editing. The toolbar on top of the window contains a navigator control (on the left) allowing to scroll the sequence display from one hit to another. The information on the currently selected hit is shown to the right of the navigator. The search field distinguishes three query types automatically: nucleotide sequence, coordinate and annotation text. The rightmost control allows to zoom the feature map in or out. This example shows the browser window in the middle of analysis of a search with Fur binding site profile. The currently selected (and highlighted in the top and middle panes) Fur binding site is located in front of a large operon on the bottom strand, most of the frames in which were originally annotated as "hypothetical protein." Searching the Uniprot and especially the Conserved Domain Database (the actual result of CDD search with one protein coded by this operon is shown) revealed that all these frames code for components of the type VI secretion system, which is known to be controlled by Fur.

might be helpful to delimit putative operons more clearly. Alternatively, the browser could be used to view an existing GenBank file independently of any search function. SigmoID allows editing of the existing features and their qualifiers and provides integrated web-based hmmsearch, hmmscan, blastn and blastp search functions for verifying the existing annotation. The search is possible against protein, nucleotide and protein domain

databases. The access to these functions is provided via the corresponding APIs (*NCBI, 2016*; *Finn, Clements & Eddy, 2011*). The genome browser can also be used to search for a sequence, a coordinate or a feature text within a genome file.

### Extracting regulon information from the annotated genome

The function "List regulons" allows listing putative transcription units controlled by either all of the regulators or by just one of them. An annotated TFBS or a promoter is considered by this function as the start of a putative transcription unit. The configuration window (Fig. S1E) allows the user to set the allowed interval for the location of promoter/TFBS relative to the first gene of a putative operon. Two cases are considered being an operon end: a non-coding gap of the specified (and user-configurable) size and a transcription terminator. The divergently transcribed neighbouring operons which might be controlled by the same TFBS could be listed as a single divergon unit. The function also outputs the counts of putative transcription units controlled by each regulator.

### Data plots

SigmoID can display simple plots of genome-related data. The "Add Plot…" command from the "Genome" menu loads the data from text files that have tab-separated pairs of genome coordinate and data value (one pair per line). It can also load the data produced by the samtools depth command (*Li, 2011*). Up to four overlapping plots can be added. This, among other possible options, allows to simultaneously display read coverage for each strand of two samples from a strand-specific RNA-Seq experiment.

RNA-Seq coverage plots could be helpful for verifying positions of regulatory sequences relatively to transcription start sites. At the moment, SigmoID does not include all functions required and can only load and display read count values. These can be produced in different ways, for example using Bowtie 2 (*Langmead & Salzberg, 2012*) and samtools as described in detail in the SigmoID help file.

## TFBS, promoter and genome data

The RegPrecise database (*Novichkov et al., 2013*) version 3 was used as the main source of data for TFBSs for different transcription factors. An integrated access to this database is provided within SigmoID via the web services API (*Novichkov et al., 2012*). The data for the regulators not present in RegPrecise (which includes all promoters) were extracted either from original publications or from RegulonDB (*Salgado et al., 2013*).

For the construction of optimised search profiles, the complete genome sequences of *Pectobacterium atrosepticum* SCRI1043 (*Bell et al., 2004*), *Pectobacterium carotovorum* PCC21 (*Park et al., 2012*), *Pectobacterium wasabiae* SCC3193 (*Koskinen et al., 2012*) and *Dickeya dadantii* 3937 (*Glasner et al., 2011*) were downloaded from the GenBank database (*Clark et al., 2016*). They were used as the main reference sequences for the soft rot *Enterobacteriaceae* group. For some profiles with few binding sites, additional genome sequences were employed (accession numbers NC_012917, CP001790, CP001836, CP001654, CP007744). Verification and calibration of *Pseudomonas*-optimised profiles were done with the genome sequences of strains referenced by RegPrecise (accession numbers AE016853, CP000076, AE004091, CT573326, CP003677, AE015451, CP000680).

## RESULTS AND DISCUSSION

### Search engine integration

Two different approaches have been routinely utilised for representation and identification of regulatory protein binding sites. Position specific scoring matrices (PSSM) are traditionally used for representing both TFBSs and promoters (*Stormo, 2000*). Profile hidden Markov models (HMM) have been often used for promoter identification (*Yada et al., 1997*; *Pedersen et al., 1996*; *Fouts et al., 2002*), but rarely applied to TFBS searches. Since we aimed at both promoter and TFBS identification, we looked for the available tools implementing both approaches. Among the open-source cross-platform tools we have chosen the HMMER and MEME Suite packages as freely available and widely used (*Eddy, 2011*; *Bailey et al., 2015*). We have found HMM-based and PSSM-based search (as implemented in nhmmer and MAST) to perform similarly in terms of specificity, sensitivity and speed (data not shown). Nevertheless, there were minor differences between the two search algorithms in detecting "weak" binding sites (with no clear advantage shown by either algorithm), therefore both nhmmer and MAST are provided as search engines within SigmoID. Subsequent filtering of search results and annotation additions can be done regardless of the search algorithm.

A similar approach has been taken for terminator identification. The actual search is performed by TransTerm HP; SigmoID implements the required sequence format conversions and adds terminator annotations to genome file in GenBank format.

The results produced by all three search engines are displayed in genome browser (Fig. 2).

### Using SigmoID to construct optimised HMM profiles

Experimental data on TF/sigma factor binding sites are available for a small fraction of the sequenced genomes. As this sort of experiments is often technically challenging and time-consuming, bioinformatics could be used to infer possible regulatory networks in related species using already available experimental data. Therefore, we aimed at collecting the regulatory information, construction of the optimised TFBS/promoter search profiles and making them easily accessible with the help of a user-friendly tool.

Before proceeding with profile construction, we verified the presence of the corresponding regulatory gene in each genome by FASTA search (*Pearson, 2004*). General profile optimisation algorithm was similar to the one used by RegPrecise (*Novichkov et al., 2010a*), but utilised nhmmer as the main search engine. For the purpose of this work, the TFBS/promoter sequences extracted from the original data source were used to build a primary HMM profile and search for similar sites in the genomes of related bacterial species. Orthologous sites found by nhmmer were always retained. Contrary to the conservative approach utilised by RegPrecise for automated regulon propagation (*Novichkov et al., 2010a*), additional conserved high scoring sites were considered and retained if located in front of the genes known or likely to be involved in the same physiological process as the characterised members of the regulog. This procedure resulted in secondary alignment composed exclusively of target genomes sequences. At this point, we evaluated the possibility to improve signal/noise ratio by verifying the boundaries of the

alignments. To do this, we applied the "Extend binding sites" function of SigmoID to add bases on either side of each sequence in the current profile, and, if additional conserved bases were discovered beyond the original borders, saved the new extended alignment (via the "Save alignment selection" function). Once the profile boundaries were determined, the target genomes were re-scanned and additional sites, if found, were added to the alignment. Finally, the profiles were calibrated to simplify their further use. In most cases the lowest nhmmer score of a specific hit matching the original data source was used for the trusted cut-off value, the highest score of a nonspecific hit—for the noise cut-off and the mean of these two values—for the gathering cut-off. The gathering cut-off is used by default in further searches with the calibrated profile.

## Calibrated profiles included with SigmoID

SigmoID includes sets of calibrated HMM profiles for two groups of bacteria. The SRE group includes plant pathogens from the *Pectobacterium* and *Dickeya* genera. Genome sequences of the two strains from this group, *Pectobacterium atrosepticum* SCRI1043 and *Dickeya dadantii* 3937, have a high-quality curated annotation (*Bell et al., 2004*; *Glasner et al., 2011*) and were used throughout this work for search and extraction of the TFBS data. To better cover sequence diversity within this group of plant pathogens, genome sequences from representatives of the other two species, *P. carotovorum* and *P. wasabiae*, were also included.

A number of transcription factors and their binding sites have been experimentally characterised in *Dickeya dadantii* while *Pectobacterium spp.* is less studied in this respect. On the other hand, the RegPrecise database (*Novichkov et al., 2013*) contains the information on TFBS for many important regulators in *P. atrosepticum* SCRI1043, which allowed us to build the corresponding search profiles. These profiles, however, performed sub-optimally when applied to other *Pectobacterium* species and especially *Dickeya*. Therefore, we have adjusted some of the profiles to perform more or less uniformly for the whole SRE group, as described above.

The profile optimisation procedure resulted in changing TFBS/promoter boundaries for the majority of profiles (27 out of 35) and more often than not increased their information content (Table 1, corresponding sequence logos are shown in Fig. S2). The cases where the information content decreased were usually the result of a small number of sequences in the original data source. Despite of the decreased information content, these profiles represent the corresponding regulog better due to the inclusion of TFBS/promoters controlling divergent regulon members.

The second set of HMM profiles was constructed for *Pseudomonas* spp. This genus is represented in RegPrecise by seven species with significant differences in genome organisation and includes plant and animal associated species, both pathogenic and non-pathogenic. Most of the characterised TFs and their binding sites appeared to be well conserved across the genus, which in many cases allowed direct use of the data present in RegPrecise for TFBS identification in *Pseudomonas* genomes. This approach appeared to be possible for the majority of TFs (25 out of 30). For only two of the TFs with data in RegPrecise (RutR and Zur) a universal profile wasn't feasible due to significant

**Table 1 Characteristics of the calibrated SRE-specific profiles included with SigmoID.[a]**

| Regulatory protein | Binding site length/ information content | | Number of binding sites used for profile construction | Source of the original binding site data |
|---|---|---|---|---|
| | Original | Final | | |
| ArcA | 15/16.9 | 19/21.0 | 33 | Ravcheev (2009) |
| ArgR | 18/17.8 | 24/23.2 | 24 | RegPrecise |
| AscG | 20/30.4 | 20/23.3 | 7 | RegPrecise |
| BaeR | 20/26.2 | 27/30.4 | 13 | Nishino, Honda & Yamaguchi (2005) |
| CpxR | 15/15.0 | 15/20.3 | 16 | RegulonDB |
| CRP | 16/10.2 | 24/15.8 | 100 | RegPrecise |
| FliA | 27/23.5 | 26/23.5 | 96 | Ide, Ikebe & Kutsukake (1999) |
| FNR | 14/12.4 | 22/20.7 | 34 | RegulonDB |
| FruR | 16/17.8 | 18/19.4 | 25 | RegPrecise |
| Fur | 19/20.2 | 23/18.8 | 98 | RegPrecise |
| GalR | 20/24.0 | 18/19.5 | 39 | RegPrecise |
| HrpL | 28/19.0 | 31/25.9 | 63 | McNally et al. (2011) |
| KdgR | 21/22.1 | 25/22.4 | 85 | Rodionov (2004) |
| LexA | 20/20.3 | 20/21.2 | 34 | RegPrecise |
| MetJ | 16/20.2 | 24/23.9 | 21 | RegPrecise |
| MetR | 15/17.3 | 21/19.2 | 18 | RegPrecise |
| NarP | 16/17.9 | 18/17.6 | 54 | RegPrecise |
| NifA | 16/14.6 | 18/19.5 | 18 | RegPrecise |
| NsrR | 19/27.9 | 19/23.2 | 20 | RegPrecise |
| NtrC | 17/22.0 | 17/17.8 | 56 | RegPrecise |
| OxyR | 17/13.3 | 15/18.1 | 19 | Seo et al. (2015) |
| PhoP1 | 19/24.4 | 16/20.4 | 24 | Harari et al. (2010) |
| PhoP2 | 19/16.4 | 16/20.3 | 31 | Harari et al. (2010) |
| PmrA | 20/23.1 | 19/22.5 | 19 | Ogasawara et al. (2012) |
| PurR | 16/18.4 | 16/17.6 | 23 | RegPrecise |
| RcsB | 14/19.6 | 18/17.4 | 39 | Andresen et al. (2010) |
| RhaS | 51/42.7 | 51/40.2 | 21 | RegPrecise |
| Rob | 17/15.6 | 20/22.7 | 45 | RegulonDB |
| RpoE | 27/18.4 | 30/20.3 | 57 | Rhodius et al. (2006) |
| RpoH1 | 28/27.5 | 33/29.6 | 20 | Nonaka et al. (2006) |
| RpoH2 | 29/24.7 | 30/30.8 | 19 | Nonaka et al. (2006) |
| RpoN | 16/14.7 | 16/21.8 | 105 | Barrios, Valderrama & Morett (1999) |
| RpoS | 26/22.8 | 35/22.9 | 25 | RegulonDB |
| SlyA | 12/10.7 | 20/19.2 | 55 | Stapleton et al. (2002), Haque et al. (2009) and McVicker et al. (2011) |
| UxuR | 18/27.6 | 22/23.7 | 17 | RegPrecise |

**Notes.**

[a]The table does not include the data on additional 9 profiles for TFs controlling just one or very few operons. These profiles are still distributed with SigmoID.

interspecies variation in regulon content and binding site composition. In this case profiles for plant-associated species were constructed.

Six *Pseudomonas*-specific profiles are based on the data not present in RegPrecise. The profiles for the virulence regulator RhpR as well as for the HrpL- and PvdS-dependent promoters were derived from experimentally determined binding sites (*Deng et al., 2010*; *Deng et al., 2014*; *Ferreira et al., 2006*; *Swingle et al., 2008*). FliA-, RpoN- and RpoH-dependent promoters appeared to be sufficiently conserved between *Enterobacteriaceae* and *Pseudomonadaceae*, which allowed us to create the corresponding promoter profiles via the standard optimisation procedure described above with SRE profiles as the starting point.

The resulting profile collection distributed with SigmoID includes 37 *Pseudomonas*-specific profiles (Table S1).

## TFBS/promoter position verification with RNA-Seq data

SigmoID can simultaneously display regulatory elements and RNA-Seq coverage graphs which helps to verify TFBS/promoter positions relative to transcription start sites. At the time of writing, one large-scale RNA-Seq experimental dataset was available from public databases for the bacterial genera discussed in this work (*Kwenda et al., 2016*). The experiment had samples prepared from *P. atrosepticum* cultures grown to stationary phase in two media. Since the difference between the samples is multifactorial, this dataset could only have limited use for TFBS/promoter verification. However, one of the alternative sigma factors, RpoS, is strongly expressed in both samples, which allowed us to define RpoS-dependent promoter profile for this bacterium.

To our knowledge, the RpoS regulon has not been characterised in SRE species. Because of this, we had difficulty constructing the HMM profile for RpoS-dependent promoters in SRE starting with *E. coli* data. nhmmer search scores were low and high uncertainty with the potential RpoS regulon members complicated selecting true hits for profile construction. The RNA-Seq data allowed us to select nhmmer hits correctly located in front of transcription start sites of the genes expressed in the stationary phase (Fig. 3A) and to construct the HMM profile for RpoS-dependent promoters in SRE. This profile has identified 24 transcriptional units as potentially controlled by RpoS in *P. atrosepticum* (Table S2). Sequence logo of their promoters (Fig. 3B) shows features typical of the RpoS-dependent promoters characterised recently for *E. coli* by ChIP-Seq analysis (*Peano et al., 2015*).

## Applying SigmoID to improving the annotation of *Pectobacterium atrosepticum* 21A genome

We have recently sequenced the genome of *P. atrosepticum* strain 21A (*Nikolaichik et al., 2014*). The version of this genome submitted to GenBank had mostly automated annotation produced by using the Prokka pipeline (*Seemann, 2014*). While testing SigmoID on this genome sequence, we noticed some inconsistencies in the annotation (known genes missing from or unlikely genes present in regulons). Therefore, we tested if SigmoID could aid to systematically improve functional annotation of this genome using all the available regulatory information. Putative terminators in this genome

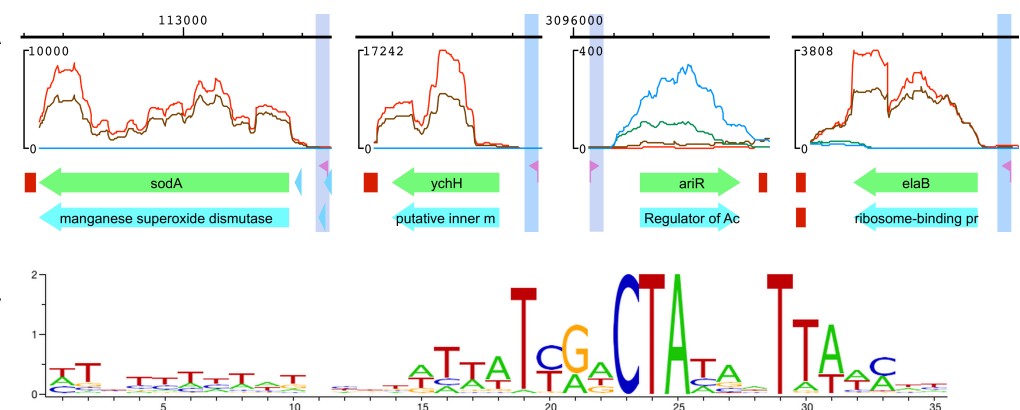

**Figure 3** **RpoS promoter verification with RNA-Seq data.** (A) Putative RpoS-dependent promoter positions relative to selected genes expressed in stationary phase. RNA-Seq reads from GEO accession GSE68547 were mapped onto *P. atrosepticum* 21A chromosome with bowtie 2 resulting in over 95% of uniquely mapped reads. Read counts were calculated with samtools, displayed by SigmoID and resulting pictures were exported as vector images. RpoS-dependent promoter is highlighted and marked by a flag sign. The counts for reads originating from the culture grown in minimal medium are represented by a blue line for the sense strand and by a brown line for the antisense strand. The counts for reads originating from the rich medium grown culture are represented by a green line for the sense strand and by a red line for the antisense strand. (B) Sequence logo of the RpoS-dependent promoter in *P. atrosepticum*.

were identified first, followed by sequential scanning of the genome using (i) all the calibrated profiles described above and (ii) uncalibrated profiles for other *P. atrosepticum* regulators present in RegPrecise. This enabled us to delimit putative transcription units controlled by known regulators. The annotation of CDS within these transcription units was verified (and edited if necessary) according to the regulatory information. At this stage the database search abilities interfaced by SigmoID within the genome browser (Fig. 2) were used extensively.

It should be noted that two independent automated annotations are available for the *P. atrosepticum* 21A chromosome, produced by the Prokka and NCBI pipelines (accession numbers CP009125.1 and NZ_CP009125.1). Both automated annotations contain errors, some of which could be easily corrected if regulatory information was taken into account. For example, three neighbouring open reading frames (locus tags GZ59_45330, GZ59_45340 and GZ59_45350) are annotated in NZ_CP009125.1 as "molecular chaperone TorD," "cytochrome C biogenesis protein CcmF" and "cytochrome C biogenesis protein CcmE." However, the inference of a CRP site in front of the first gene suggests that the operon might be related to the catabolism of secondary carbon sources. The presence of the AscG binding site near the CRP site gives a hint as to what such a source could be, as AscG is a specialised regulator of beta-glucoside utilisation genes (*Ishida, Kori & Ishihama, 2009*). Searching the Conserved Domain Database shows a strong similarity of the protein products coded for by the GZ59_45330, GZ59_45340 and GZ59_45350 loci to PTS components. The results of NCBI non-redundant protein database searches, however, contain a mixture of PTS components and unrelated TorD/CcmF/CcmE hits. A direct check of homology between GZ59_45330/GZ59_45340/GZ59_45350 and experimentally characterised TorD/CcmF/CcmE from *E. coli* shows no significant similarity. Further
analysis of NZ_CP009125.1 revealed that TorD and CcmF were present in the annotation four times and CcmE—three times, and in all cases except two (one copy each of CcmE and CcmF) these annotations were incorrect: these frames should be annotated as cellobiose-specific PTS system subunits IIA, IIC and IIB.

Although most often the errors were only present in one of the annotations, in some cases both annotations were either incorrect or significantly incomplete. Table S3 gives some of the examples where these errors were noticed and corrected with the regulatory information taken into account.

Once the *P. atosepticum* 21A genome annotation has been updated, we have used the "List regulons" function of SigmoID for the final check and resolved few issues with missing or inconsistent annotation within putative transcription units.

The updated annotation of *P. atrosepticum* 21A genome (accession number CP009125.1) contains about three hundred changes in gene/CDS features introduced in accordance with regulatory information (Table 2). Most of the changes replace a "hypothetical protein" with a more meaningful product name. Other changes, however, improve incomplete annotation or replace previous versions entirely.

Overall, the SigmoID-assisted review of *P. atosepticum* 21A genome annotation adds 930 transcription/sigma factor binding sites for 74 regulatory proteins and over 1,000 transcriptional terminators. A total of 592 putative transcription units with identified regulator binding sites suggestive of their regulation and physiological role (Data S1) include approximately 40 percent of the genes in the *P. atosepticum* 21A chromosome.

## CONCLUSIONS

The growing accessibility of genomic sequencing is opening up genome sequence analysis to more and more bench scientists, and not all of them are comfortable with the command line tools. SigmoID was designed to speed up extraction of regulatory information from genomic sequences and databases, present it visually and to provide an easy and convenient access to important command line search tools for a larger number of biologists.

Binding of a regulatory protein to DNA is rarely studied in more than one or very few species. DNA binding sites of orthologous proteins may diverge significantly, sometimes even between the genera of the same family. Hence, a TFBS profile built with the data for one taxonomic group will perform sub-optimally in a more or less distant species that has orthologous regulator. SigmoID provides helpful utilities for optimising TFBS/promoter profiles for a new taxonomic group. This allows to get a better understanding of transcriptional regulation in a wider range of bacteria.

Graphical display of regulatory information in genomic context suggests possible operon structure/regulation and highlights annotation problems while database search and annotation editing possibilities of SigmoID in many cases allow correcting these problems. Therefore, we hope that SigmoID might be a useful complement to automated annotation pipelines at the final stages of genome annotation.

There are other potential uses for SigmoID that are beyond the scope of this paper. The current SigmoID distribution contains two large collections of optimised TFBS/promoter

**Table 2   Summary of the updates made to the annotation of *P. atrosepticum* 21A genome by using SigmoID.**

| Regulatory protein | Binding sites found | Edited downstream loci | | |
|---|---|---|---|---|
| | | CDS feature changed | Gene name changed | Total edited loci |
| ArcA | 31 | 5 | 4 | 8 |
| AscG | 4 | 3 | 6 | 7 |
| BaeR | 3 | 1 | 1 | 1 |
| CpxR | 35 | 5 | 9 | 9 |
| CRP | 110 | 28 | 31 | 44 |
| DcuR | 10 | 1 | 3 | 3 |
| FliA | 33 | 8 | 7 | 8 |
| FNR | 92 | 23 | 24 | 29 |
| FruR | 14 | 5 | 5 | 5 |
| Fur | 85 | 45 | 36 | 50 |
| GalR | 10 | 3 | 0 | 3 |
| HrpL | 6 | 5 | 3 | 5 |
| KdgR | 43 | 10 | 8 | 13 |
| LexA | 20 | 1 | 4 | 4 |
| MetJ | 14 | 4 | 6 | 6 |
| NarP | 22 | 2 | 5 | 5 |
| NsrR | 12 | 1 | 1 | 1 |
| OxyR | 13 | 3 | 1 | 3 |
| PhoB | 10 | 0 | 2 | 2 |
| PhoP | 24 | 3 | 6 | 6 |
| PurR | 12 | 1 | 2 | 2 |
| RcsB | 24 | 4 | 6 | 6 |
| Rob | 25 | 13 | 8 | 13 |
| RpoE | 19 | 6 | 2 | 8 |
| RpoH | 9 | 0 | 3 | 3 |
| RpoN | 38 | 32 | 22 | 34 |
| RpoS | 24 | 9 | 9 | 10 |
| SlyA | 22 | 13 | 11 | 14 |
| UxuR | 5 | 0 | 4 | 4 |
| Zur | 4 | 3 | 1 | 3 |

profiles that may be useful in studies of transcriptional regulation in SRE and *Pseudomonas*. Importantly, this work resulted in constructing optimised profiles for 41 transcriptional regulator for *Pectobacterium* genus. Some of these regulators have little or no experimental data in pectobacteria (ArcA, BaeR, CpxR, OxyR, PhoB, PhoP, PmrA, RpoS, Rob, SlyA). We also propose profiles for three regulators (HrpS, VasH and RtcR) with unknown binding sites. A preliminary assessment of the inferred regulons controlled by Rob, SlyA, PhoP and other regulators implies their importance for pathogenicity and should facilitate experimental characterisation of their role in host-pathogen interaction.

## ACKNOWLEDGEMENTS

We thank Alexander Nikolaichik for advice on compiling the HMMER package for Windows and Sveta Makovets for critical reading of the manuscript and helpful suggestions.

### Funding

This work was supported in part by the State Research Programme ''Biotechnology'' within projects 2.52 and 2.24. The funders had no role in study design, data collection and analysis, decision to publish, or preparation of the manuscript.

### Grant Disclosures

The following grant information was disclosed by the authors:
The State Research Programme ''Biotechnology''.

### Competing Interests

The authors declare there are no competing interests.

### Author Contributions

- Yevgeny Nikolaichik conceived and designed the experiments, performed the experiments, analyzed the data, contributed reagents/materials/analysis tools, wrote the paper, prepared figures and/or tables, reviewed drafts of the paper.
- Aliaksandr U. Damienikan performed the experiments, analyzed the data, contributed reagents/materials/analysis tools, prepared figures and/or tables, reviewed drafts of the paper.

### DNA Deposition

The following information was supplied regarding the deposition of DNA sequences:
GenBank accession number CP009125.1.

### Data Availability

Data is available at Github:
https://github.com/nikolaichik/SigmoID.

### Supplemental Information

Supplemental information for this article can be found online at http://dx.doi.org/10.7717/peerj.2056#supplemental-information.

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
