# Peer review of "SigmoID: a user-friendly tool for improving bacterial genome annotation through analysis of transcription control signals"

_PeerJ, doi:10.7717/peerj.2056_

## Round 0.1 · original submission · Major Revisions

· Academic Editor

Major Revisions

As Reviewer 2 pointed out, I ask you to rewrite the article. In its current form the title is misleading, since the major part of your article is concered with the annotation of one specific species only. The central topic of the article should be concerned with your method / software. I am actually skeptical how much your software is in fact a general purpose software rather than tuned for the organism that you analysed. As you point out in the introduction, see lines 87-97, the readers gets the impression that one main purpose of SigmoID is to analyse genomes from the Pectobacteria genus. This is then continued in the Methods and Data section. Thus in order to reconsider your article for publication in PeerJ, you need to change the focus as described by Reviewer 2. Thus, I ask you to add another typical more generic use case for your software.

·

Basic reporting

No Comments

Experimental design

No Comments

Validity of the findings

No Comments

Comments for the author

The manuscript "SigmoID: a user-friendly tool for improving bacterial genome annotation through analysis of transcription control signals" describes a graphical user interface based wrapper for the widely used sequence analysis tools HMMER, MEME, TransTermHP. The described open source tool SigmoID was mainly created to make the prediction of bacterial transcription factors more accessible to researchers without strong bioinformatical skills and to extend genome annotations in Genbank format by the search results. The manuscript also describes the application of the tool to the analysis of the genome of the plant pathogen Pectobacterium atrosepticum.

I was able to run the tool on a Ubuntu GNU/Linux 15.10 system and perform predictions as described in the manuscript.

1) While the main aim of the tool is to extend available Genbank files it might be useful to offer the user that the sequence search hits are stored in a separate file (maybe in the original format).

2) Regarding coding style and the git repository:
* Part of the code contains rather long functions/methods/lines and the coding style of the scripts was partially not pythonic (e.g. "if enter.palindromic is True:" should be "if enter.palindromic:"). Please see https://www.python.org/dev/peps/pep-0008/
* I would suggest to have dedicated subfolder for the Python script inside the repository. This would declutter the root of the repository.
* To ensure long term access to the code deposition at repositories like zenodo or figshare is recommended. Zenodo offers webhooks to automatically store new releases of github repositories. See https://guides.github.com/activities/citable-code/.

3) The authors write: 'To do this, we applied the "Extend binding sites" function of SigmoID, which adds ten bases on either side of each sequence in the current profile [...]'. Why are ten bases added? I think it would be helpful if the user can choose the number of bases for the extension.

·

Basic reporting

The authors of the article entitled
“SigmoID: a user-friendly tool for improving bacterial genome annotation through analysis of transcription control signals”
present an automated tool to annotate protein-coding genes in bacterial genomes with transcriptional signals, i.e. transcription factor binding sites (TFBS) and transcription terminator signals. Their framework integrates a genome browser that allows for a visual inspection of the results. The authors demonstrate how their tool can be applied to improve the annotation of bacterial genomes. In particular they apply their tool to improve the annotation of Pectobacterium atrosepticum. With this they provide improved annotations for about 300 elements.

The authors appropriately describe their tool as making the respective analyses easier without providing novel algorithms. By this the integrated tools such as HMMER, MEME and TransTermHP are made accessible for non-experts.
The software is made available as open source.

The following points should be addressed:

In general the manuscript should be more balanced towards the description of the software. The title of the manuscript as well as the abstract put the primary focus on the software with presenting the application to Pectobacterium atrosepticum as an example.
However, in the manuscript body a strong focus is set on the analysis of the P. atrosepticum genome annotation.

In particular I suggest the following:

The description of the single functionalities of the software should be extended in some cases.

E.g. the ‘Transcription terminator search’ is basically described in one sentence. TransTermHP is a program with a complex algorithm and many parameters. Are default parameters used in any case? Are parameters automatically optimised? Can the user set some of the main parameters?

Also the section on ‘Extracting operon and regulon information’ should be extended. SigmoID seems to implement an algorithm by which TFBS and terminators are combined in order to predict operons. This is actually a non-trivial problem when considering that operons can contain conditional internal promoters and terminator signals. A more detailed description of the approach would be good.

In addition I suggest, to remove any reference to Pectobacteria from the methods sections if possible. E.g. in ‘Genome scan’.

The application of SigmoID to P. atrosepticum is described in much detail in the two sections ‘Using SigmoID to construct hmm profiles optimised for pectobacteria’ and ‘Applying SigmoID to improving the annotation of Pectobacterium atrosepticum 21A genome’. These sections are very long and have no clear sub-structure. In order to improve the readability for the non-expert I suggest to introduce subsections for each transcription factor or set of genes that is discussed. In addition, I suggest to keep each of these subsections as concise as possible.

Furthermore, the following minor points should be addressed:

In the introduction the authors mention the time and workload as disadvantages of manual genome annotation. I suggest to also mention reproducibility as an advantage of automated methods.

In the section ‘TFBS, promoter and genome data’ the authors list a number of genome sequences that were retrieved as well as ‘a few other sequences’. Please specify.

The authors state that ‘SigmoID processes both nhmmer and MAST results to remove the likely false positive hits’. Does this mean that the intersection of both methods is used?

Finally, I suggest that the manuscript should be proof-read by a native speaker if possible.

Experimental design

No Comments

Validity of the findings

No Comments

---

## Round 0.2 · Minor Revisions

· Academic Editor

Minor Revisions

Dear authors,

please correct the minor revision details pointed out by Reviewer 1 (Konrad Förstner). Thank you.

·

Basic reporting

No Comments

Experimental design

No Comments

Validity of the findings

No Comments

Comments for the author

The authors have addressed the raised points adequately. The major rewriting improved the manuscript in my opinion. Unfortunately, it also introduced several new errors and I would highly recommend to have another round of proofreading. For example the link to the GitHub repository is missing now while the text states "The compiled applications for all three platforms are also available from the same GitHub repository". Just add the GitHub link again. Another example is "GPL 3.0 licence ()" which is missing the content inside of the brackets. There are several other errors like this that should be corrected before acceptance.

·

Basic reporting

The authors of the article entitled “SigmoID: a user-friendly tool for improving bacterial genome annotation through analysis of transcription control signals” provide a revised version of their manuscript. The description of the software as well as the structure of the manuscript have been improved significantly. Furthermore, the authors supplemented the presentation of the software with additional figures.

Experimental design

No Comments

Validity of the findings

No Comments

---

## Round 0.3 · accepted · Accept

· Academic Editor

Accept

Dear authors,

Many thanks for your revised version and I have to apologize for the possible problems between the two revised versions. I am currently not sure where the problem really arose, but I am relieved that you could solve this so quickly.